
# Chemical Composition of PM2.5 in October 2017 Northern California Wildfire Plumes

Yutong Liang[1], Coty N. Jen[1,2], Robert J. Weber[1], Pawel K. Misztal[1,3], Allen H. Goldstein[1,4]

[1]Department of Environmental Science, Policy, and Management, University of California, Berkeley, Berkeley, CA, 94720, USA
[2]Department of Chemical Engineering, Carnegie Mellon University, Pittsburgh, PA 15213, USA
[3]Department of Civil, Architectural and Environmental Engineering, University of Texas at Austin, Austin, TX 78712, USA
[4]Department of Civil and Environmental Engineering, University of California, Berkeley, California 94720, USA

*Correspondence to*: Yutong Liang (yutong.liang@berkeley.edu)

**Abstract.** Wildfires have become more common and intense in the western US over recent decades due to a combination of historical land management and warming climate. Emissions from large scale fires now frequently affect populated regions such as the San Francisco Bay Area during the fall wildfire season, with documented impacts of the resulting particulate matter on human health. Health impacts of exposure to wildfire emissions depend on the chemical composition of particulate matter, but the molecular composition of the real biomass burning organic aerosol (BBOA) that reaches large population centers remains insufficiently characterized. We took PM2.5 (particles having aerodynamic diameters less than or equal to 2.5 µm) samples at the University of California, Berkeley campus (~60 km downwind of the fires) during the October 2017 Northern California wildfires period, and analyzed molecular composition of OA using a two-dimensional gas-chromatography coupled with high resolution time-of-flight mass spectrometer (GC×GC ToFMS). Sugar-like compounds were the most abundant component of BBOA, followed by mono-carboxylic acids, aromatic compounds, other oxygenated compounds and terpenoids. The vast majority of compounds detected in smoke have unknown health impacts.

Regression models were trained to predict the saturation vapor pressure and averaged carbon oxidation state ($\overline{OS_c}$) of compounds. The compounds speciated have a wide volatility distribution and most of them are highly oxygenated. In addition, time series of primary BBOA tracers observed in Berkeley were found to be indicative of the types of plants in the ecosystems burned in Napa and Sonoma, and could be used to differentiate the regions from which the smoke must have originated. Commonly used secondary BBOA markers like 4-nitrocatechol were enhanced when plumes aged, but their very fast formation caused them to have similar temporal variation as primary BBOA tracers. Using hierarchical clustering analysis, we classified compounds into 7 factors indicative of their sources and transformation processes, identifying a unique daytime secondary BBOA factor. Chemicals associated with this factor include multifunctional acids and oxygenated aromatic compounds. These compounds have high $\overline{OS_c}$, are also semivolatile. We observed no net particle-phase organic carbon formation, which indicates an approximate balance between the mass of evaporated primary and secondary organic carbonaceous compounds to the addition of secondary organic carbonaceous compounds.



# 1 Introduction

Biomass Burning (BB) is the largest source of carbonaceous aerosols and the second largest source of non-methane organic gases in Earth's atmosphere (Akagi et al., 2011; Bond et al., 2004). Driven by forest management practices (such as fire suppression) and climate change, the size and frequency of wildfires in the western United States have been steadily increasing over the past 20 years (Abatzoglou and Williams, 2016; Dennison et al., 2014; Westerling et al., 2006), which results in worsening air quality in this region (McClure and Jaffe, 2018).

Organic aerosol particles (OA) are the main component of particulate matter (PM) emitted in biomass burning (Fine et al., 2004; Nolte et al., 2001). Many gas phase volatile organic compounds (VOCs), intermediate and semi-volatile organic compounds (I/SVOCs) are also emitted in biomass fires (Grieshop et al., 2009a, 2009b; May et al., 2013). The oxidation and condensation of gas phase organics, in addition to the evaporation of particulate organics, all affect the mass and composition of biomass burning organic aerosol (BBOA) (Hodshire et al., 2019a). Numerous studies have connected human health outcomes with exposure to wildfire smokes (Reid et al., 2016; Sigsgaard et al., 2015). BBOA was shown to be the most toxic component of water soluble $PM_{2.5}$ (particles having aerodynamic diameters less than or equal to 2.5 μm) in the Southeastern United States, in terms of the ability to generate reactive oxygen species (which can cause cell damage) per unit mass (Verma et al., 2015). The chemical composition of BBOA, which is related to the fuel, combustion condition and atmospheric aging, affects the potential toxicity of BBOA. On per mass of $PM_{2.5}$ basis, aerosols generated in flaming combustion have stronger mutagenicity and lung toxicity potencies than smoldering BBOA (Kim et al., 2018). A recent study shows atmospheric aging for two days can increase the oxidative potential (the ability of particle to generate reactive oxygen species) of BBOA by a factor of $2.1 \pm 0.9$ (Wong et al., 2019).

The chemical composition of particle-phase BBOA emissions has been comprehensively characterized in laboratory burning experiments, e.g., Hays et al. (2002), Jen et al. (2019) and Simoneit et al. (1993). These studies provide extensive inventories of BBOA emissions and discovered many chemical tracer compounds for different vegetation types. However, there is evidence showing the emission of BBOA in wildfires are different from that in the laboratory settings. BBOA emissions from natural fires depend on the moisture of the fuels and many environmental factors such as wind and temperature, which is hard to mimic in laboratory fires (Andreae, 2019). The aging processes of BBOA have also been studied in many laboratory oxidation experiments, such as Bertrand et al. (2018), Fortenberry et al. (2018) and Sengupta et al. (2020). These studies provide substantial information on the oxidation mechanisms, and identified many potential marker compounds for biomass burning secondary organic aerosol (BB SOA). However, the oxidation environment in chambers or flow reactors are also different from the wildfires. Variabilities in dilution and levels of background aerosols, in addition to the variabilities in emission and chemistry, all affect OA evaporation and BB SOA formation (Hodshire et al., 2019b). In laboratory studies, aging was often shown to increase the mass of BBOA. However, increase of BBOA mass within aging



plumes was only observed in a few field studies (Hodshire et al., 2019a). Speciated measurements enable us to examine whether primary BBOA marker compounds emitted in fires that differ in fuel type, intensity, or other factors, are suitable for

use in source apportionment, and to identify useful marker compounds (and formation pathways) for BB SOA. Speciation of aged wildfire BBOA also informs us about the compounds people were exposed to during the fires.

Traditional speciated measurements of BBOA typically rely on the gas chromatography-mass spectrometric (GC-MS) analyses of filter samples. Due to the limited capability of conventional GC-MS in separating compounds, a high fraction of

BBOA is typically assigned as unresolved complex mixture (UCM) (Fine et al., 2004; Hays et al., 2004). For ambient BBOA measurements, low sampling frequency (e.g. one or two samples per day) typically makes it challenging to distinguish BBOA from other pollution sources and to capture short timescale variability indicative of changing emissions, transport, and atmospheric processing. The advances in analytical instruments, such as the two-dimensional GC coupled with high resolution mass spectrometry (GC × GC HRMS) significantly expands the ability to separate, identify and quantify organic

compounds in aerosols (Alam and Harrison, 2016; Laskin et al., 2018; Liu and Phillips, 1991; Worton et al., 2017). Using this approach, an unprecedented number of compounds in OA can be now speciated and quantified, and traced back to sources (Zhang et al., 2018). Here we apply GC × GC HRMS to the analysis of 3-4 hour time-resolution samples of extreme levels of ambient particulate matter generated by distant wildfires affecting a populated region.

In October 2017, a series of wildfires took place in Napa and Sonoma Counties in Northern California. The emissions caused extreme air pollution conditions with poor visibility throughout the highly populated San Francisco Bay Area for more than 10 days. According to US EPA's measurement data (EPA AirNow), the levels of $PM_{2.5}$ exceeded 100 μg m$^{-3}$ at multiple measurement sites within the Bay Area. A recent epidemiological study shows exposure to wildfire smoke in California during 2015-2017, including the 2017 fires measured here, significantly increased the risk of out-of-hospital cardiac arrest

(odds ratio 1.70, 95% confidence interval 1.18-2.13) (Jones et al., 2020). These dense smoke periods provide a unique chance to study the speciated chemical composition and transformations of BBOA occurring during the same time health effects have been documented. When these fires first began, we set up a comprehensive set of online measurements at the University of California, Berkeley (UCB) campus, and collected particulate filter samples for offline analysis. This article focuses on the organic aerosols collected on the filters. Main objectives for this study include (1) measuring the chemical

composition of BBOA after several hours of atmospheric aging and estimating the volatility and the average carbon oxidation state distribution of BBOA; (2) testing whether marker compounds can be used to distinguish the fuels burned in the fires; (3) testing whether traditional BBSOA tracers can indicate the aging process; (4) using a statistical approach to cluster all the compounds by their temporal behavior to examine their sources and transformation processes; and (5) identifying BBSOA marker compounds, and exploring the temporal trend of BBSOA based on the clustering analysis.



## 2 Methods

### 2.1 Fires, measurement site, characterization of backward trajectories calculation

The 2017 Northern California wildfires started in Napa and Sonoma Counties on October 8 under the influence of the hot, dry, and strong Diablo winds, and continued for more than two weeks. The northeasterly wind carried the smoke to the San Francisco Bay Area creating multiple separate events of high particulate matter concentrations and human exposure. Major fires include the Atlas Fire (size: 209 km$^2$) in Napa County; and the Tubbs Fire (149 km$^2$), the Nuns Fire (220 km$^2$), and the Pocket Fire (70 km$^2$) in Sonoma County. Emissions from such large wildfires have rarely been extensively measured in previous field campaigns (Hodshire et al., 2019a). Descriptions of these fires can be found on the Cal Fire website: https://www.fire.ca.gov/incidents/2017. A postfire survey by the Sonoma County Agricultural Preservation and Open Space District comprehensively measured the vegetation burned in the Sonoma County fires (http://sonomavegmap.org/data-downloads/). The canopy damage data obtained in that survey was used to estimate the quantity of fuels burned in the Sonoma County Fires. For the Atlas Fire, we overlaid Cal Fire's fire perimeter map and compared with a vegetation map in 2004 to estimate the vegetation burned (Thorne et al., 2004). The NASA Near Real-Time VNP14IMGTDL_NRT VIIRS 375 m Active Fire Detections product (https://earthdata.nasa.gov/active-fire-data) was used to confirm the daily fire points during the sampling period (Schroeder et al., 2014). Geospatial calculations were performed with the Geoprocessing Tools in ArcGIS Pro.

Our measurements started on October 9, on a balcony on the 2$^{nd}$ floor of a 3-story building on the UCB campus (37.873N, 122.263W), in an urban area approximately 55-65 km downwind of those fires. The Hybrid Particle Lagrangian Integrated Trajectory (HYSPLIT) model was used to calculate 24-h backward trajectories from our campus at a height of 50 m above the ground every hour in the sampling period, using Global Data Assimilation System's 0.5° × 0.5° resolution product as the meteorological input (Stein et al., 2015). Backward trajectories were grouped using the clustering function in the HYSPLIT software. The three-cluster solution was chosen.

### 2.2 PM$_{2.5}$ sampling and filter analysis by GC×GC

A total of 74 time-resolved (3-4-hour resolution) particulate matter samples were collected using our custom-made sequential filter sampler. Ambient air passed through a sharp cut 2.5 µm cyclone at 21 lpm and the particulate matter was collected on 102 mm diameter quartz-fiber filters (Pallflex Tissuquartz). The filters were stored in a -20°C freezer before analysis. One or two 0.41 cm$^2$ punches of sample were taken from each 102 mm sampled filter. Isotopically labeled internal standards were added to the filter punches prior to analysis to correct for any drift in system response (Jen et al., 2019). The samples were thermally desorbed in Helium at 320°C in a thermal-desorption autosampler (Gerstel) with online derivatization by MSTFA (n-methyl-n-trimethylsilyltrifluoroacetamide, Sigma Aldrich) before being focused on the Gerstel glass wool cooled injection system (30°C). The derivatization step converted hydroxy groups into trimethylsilyl (TMS)





esters, making the compounds easier to elute from columns. The analytes were then quickly injected into the GC system (Agilent 7890). The first column in the GC×GC system is a semi-polar capillary 60 m × 0.25 mm × 0.25 μm column (Rxi-5Sil, Restek), which mainly separates compounds by volatility. The temperature of the first GC column was programed to ramp from 40 to 320°C at 3.5°C/min, and to hold for 5 min at 320°C with helium carrier gas flowing at 2 mL min$^{-1}$. Analytes eluted from the first column were focused on a guard column (Restek, 1.5m × 0.25 mm, Siltek) in a dual-stage thermal modulator (Zoex) and then loaded onto the second column (Restek, Rtx-200MS, 1m × 0.25mm × 250 μm), where the analytes were separated mainly by polarity. A Tofwerk high resolution (m/Δm ≈ 4000) time-of-flight mass spectrometer (HR-ToF-MS) was used as the detector. All samples were analyzed under electron impact ionization (70 eV). Selected samples were analyzed by Vacuum Ultra-Violet (VUV) ionization (10.5 eV) provided by Beamline 9.0.2 at the Advanced Light Source, Lawrence Berkeley National Laboratory. The VUV analyses were for identification only. To minimize fragmentation in VUV, the temperature in the ionization chamber were maintained at 170 °C instead of 270 °C which was used for EI analysis (Isaacman et al., 2012). The GC chromatograms were analyzed using GC Image software (GC Image, LLC).

## 2.4 Compound identification, classification, and quantification

We first identified compounds in the samples by matching with authentic standards. A custom-made biomass burning standard mixture of 99 compounds including alkanes, acids, sugars, aromatic compounds, and polycyclic aromatic hydrocarbons (PAHs) was injected onto blank filters and analyzed by the same instrument. The standard compound list has been published in Table S4 in Jen et al. (2019). For compounds not in this standard mixture, we used the NIST MSSEARCH software to compare them with entries in NIST-14, MassBank, Golm Metabolome Database (GMD), Adams Essential Oil, MANE2010 flavor and fragrance mass spectral databases; and the GoAmazon, SOAS and FIREX mass spectral libraries created at UC Berkeley in previous studies using the same instrument as this study (Jen et al., 2019; Yee et al., 2018; Zhang et al., 2018). Linear retention index (RI) on the 1$^{st}$ dimension describes the elution order of compounds from the first column (Yee et al., 2018). For compounds analyzed by the same type of column, the elution order is expected to be the same. The RI match is considered in the matching process. The parent ions of the compounds were confirmed with the VUV mass spectra. Positive identifications were achieved for 43% of the speciated compounds. Details for compound identification are discussed in the Supplement.

A total of 572 compounds separated by the GC×GC were classified based on functionality, into mono-carboxylic acid (acid hereafter), alcohol, alkane, aromatic (mono-cyclic only), nitrogen-containing, other oxygenated (with 2 or more -OH or -COOH groups), (substituted/oxygenated) PAH, sugar (and sugar derivatives including anhydro-sugars and sugar alcohols), (di-/tri)terpenoid and unknown groups. Detailed procedures for classing unidentified compounds are provided in the Supplement.





The compound quantification procedure applied has been documented in detail in the main text and supplement of Jen et al. (2019). In brief, we injected multiple known levels of the 99-compound standard mix with the internal standard mix to blank filters and obtained the response curve (based on total ion count) for each compound. Sample compounds within this list were quantified using these curves. Compounds not in this standard mix were quantified using the response curve of the nearest standard compound (preferably in the same class with the compound being quantified) on the GC×GC space. As estimated by Jen et al. (2019), compounds exactly matched with a standard compound have an uncertainty ~ ±10%. Compounds quantified by the nearest compound in the same class have an uncertainty of ~ ±30%. Compounds with unknown functionality have a systematic uncertainty of 200%. We expect compounds with second column retention time > 1.6 s to also have such high uncertainty because there were no standard compounds with that high polarity, and a surrogate standard with lower polarity was used for quantification. However, only 7 reported compounds were in that chromatographic region with extremely high quantitative uncertainty.

## 2.4 Supporting measurements

Organic carbon (OC) and elemental carbon (EC) of punched samples from the filters were analyzed on a Sunset Laboratory Model 5 OC/EC Aerosol Analyzer with the NIOSH870 thermal protocol at the Air Quality Research Center at the University of California, Davis. A proton transfer reaction time of flight mass spectrometer (PTR-ToF-MS, PTR-TOF 8000, IONICON Analytik GmbH) was used to measure time-resolved concentrations of acetonitrile (a stable tracer for biomass burning) and other VOCs at a 1 Hz sampling frequency. The instrument was calibrated with an authentic VOC gas standard mixture (Apel Riemer Environmental Inc., Miami, FL) containing 23 compounds spanning a wide range of m/z. For compounds not directly calibrated, sensitivity factors derived from known proton transfer rates (Cappellin et al., 2012; Pagonis et al., 2019) and detector transmission were used to convert the response (normalized count rates) to concentration (ppb). Details of the PTR-TOF-MS operation and data processing have been documented elsewhere (Liu et al., 2019; Tang et al., 2016). Hourly concentrations of carbon monoxide (CO) and PM$_{2.5}$ were continuously measured by the Bay Area Air Quality Management District (BAAQMD) at various sites in the region. Solar radiation data measured at Bethel Island (between the fire sites and Berkeley) were also provided by BAAQMD.

## 2.5 Estimation of compound's volatility and average carbon oxidation state ($\overline{OS_c}$) from GC×GC measurement

Volatility (effective saturation concentration) and average carbon oxidation state distribution are important parameters for predicting the chemistry of OA (Donahue et al., 2011, 2012; Kroll et al., 2011). Isaacman et al. (2011) showed that the effective saturation concentrations (C*) and oxygen-to-carbon ratios of compounds can be estimated by their two-dimensional retention times. However, that model does not work as well when the sample contains a mixture of aliphatic and aromatic compounds. Also, derivatization can affect the saturation vapor pressure of compounds. We therefore trained two regression models using the MATLAB (version 2019b) Regression Learner to predict the saturation vapor pressure $v_P$ of





every compound, and the $\overline{OS_c}$ for compounds with unknown formulae. Compound class, first column retention index and second column retention time were chosen as the inputs. The compound class is related to the number of derivatized groups and thus corrects for the effect of derivatization on the retention times. We first found the standard compounds and a few positively identified oxygenated aliphatic compounds (such as malic acid, 2-hydroxyglutaric acid, and pinic acid) on the 2D

chromatogram and classified them using the same method for classifying compounds in the samples. The MPBPWIN v1.44 component (modified Grain method) in the EPI Suite and the EVAPORATION model were used to retrieve or calculate the saturation vapor pressure of these compounds at 298 K (Compernolle et al., 2011; US EPA, 2012). The average carbon oxidation state of each compound was calculated by $\overline{OS_c} = 2 \times \frac{O}{C} - \frac{H}{C} - 5 \times \frac{N}{C}$, where O, C, H and N are the numbers of oxygen, carbon, hydrogen, nitrogen (+V oxidation state) atoms in this compound, respectively. Then we trained the models

using the retention times/indices, classes of the known compounds as the inputs, and their saturation vapor pressure from databases or $\overline{OS_c}$ as responses with 5-fold cross-validation. The saturation vapor pressure model and the $\overline{OS_c}$ model achieved an $R^2$ of 0.93 and 0.96 between the modeled responses and the true responses, respectively (Figure S2). The uncertainty for the predictions of compounds in different chemical classes were also shown in Figure S2. More details about the models can be found in the Supplement. The saturation vapor pressure $v_P$ of each compound was then converted to C* (in µg m⁻³) by:

$$C_i^* = \frac{MW_i 10^6 \xi_i v_{P,i}}{760RT} \tag{1}$$

where $MW_i$ is the molecular weight of compound $i$ in g mol⁻¹ (assume MW = 200 g mol⁻¹ for compounds with unknown formulae), $\xi_i$ is the unitless activity coefficient of compound $i$ (assumed to be 1), $v_{P,i}$ is the saturation vapor pressure of compound $i$ (Torr), R is the gas constant (8.21 × 10⁻⁵ m³ atm mol⁻¹ K⁻¹) and T is the temperature (assume 298 K) (Isaacman et al., 2011; Pankow, 1994).

**2.6 Hierarchical clustering analysis (HCA)**

Agglomerative hierarchical clustering analysis (HCA) was performed to group the compounds into factors based on their temporal behaviors, using MATLAB Statistics and Machine Learning Toolbox. It has been demonstrated that HCA can identify major groups of compounds (ions) from timelines and patterns of behaviors from chamber measurement data (Koss et al., 2020). A major advantage of the hierarchical clustering analysis over the positive matrix factorization (PMF) method

is each compound will only end up in one factor. Ubiquitous biomass burning tracers like levoglucosan will not be split into multiple biomass burning factors.

The concentration timelines were first normalized to prevent all the high (or low) concentration compounds get clustered into the same factor. Then the Euclidean distance between each pair of normalized timelines (e.g. compound concentration

vectors $u$ and $v$) is calculated by:

$$d_{uv} = \sqrt{(u_1 - v_1)^2 + (u_2 - v_2)^2 + ... + (u_{74} - v_{74})^2} \tag{2}$$





where $u_i$ and $v_i$ are the normalized concentrations of compound u and v at time step $i$, respectively. The Ward's method was used to cluster the normalized timeline according to the distance (Ward, 1963). This algorithm starts with one compound as a cluster of its own, and then find the nearest compound and merge them (for example, compound $u$ and $v$ were merged in to

cluster A). When two clusters A and B are merged, the increase of within-cluster sum of squares is calculated by:

$$d(A,B) = \sqrt{\frac{2n_A n_B}{(n_A + n_B)}} \parallel \overline{A} - \overline{B} \parallel \qquad (3)$$

where $n_A$ and $n_B$ are the number of compounds in cluster A and B, $\parallel \overline{A} - \overline{B} \parallel$ is the Euclidean distance between the center of cluster A and cluster B. The goal is to find B that minimizes $d(A,B)$. The number of clusters were set at 4-8, and the 7-cluster solution was chosen mainly because of interpretability. The cost of merging (increase in $d$) was also considered by making

sure there was not a jump in $d$ when an extra merge was performed.

## 3 Results and discussion

### 3.1 Fires and fuels, backward trajectories and spread of the fire plumes

Figure 1a displays the perimeters of the wildfires in Napa and Sonoma Counties. Figure 1b shows the satellite image and the fire points detected by VIIRS on Oct 12 as an example. The UCB campus site, and many regions in the Bay Area were

directly affected by the smoke transported to the Bay Area from these fires. Fuels burned in the Atlas Fire (Fig. 1c) were dominated by hardwood (including grapevines, various oaks and eucalyptus) and shrubs (chamise and white-leaf manzanita). Conifer vegetation only accounted for 0.3% of the area within the perimeter of the Atlas Fire. Hardwood forest (various oaks) also dominated the canopies burned in fires in the Sonoma County. However, conifers (Douglas fir, knobcone pine, redwood, ponderosa pine, etc.) contributed 20.9% of the canopy burned. The shrubs (shrubby oaks, chamise, manzanita)

constitute 8.9% of canopy burned in the Sonoma County (Fig. 1d).

The mean backward trajectory of each cluster is also shown on Fig. 1a. Plumes in cluster 1 arrived from the northeast at relatively low speed. They mainly picked up smoke from the Atlas Fire. Plumes in cluster 2 originated from the west coast. They picked up smoke from the wildfires (mainly Sonoma County fires) and then transported it to the Bay Area. Smoke in

cluster 2 is expected to be more aged than smoke in cluster 3. Plumes in cluster 3 traveled 3-5 hours from the fires to the UCB campus, as estimated from HYSPLIT. They mainly transported smoke from the Sonoma County Fires to the Bay Area.

Figure 2 shows the impact of three representative wildfires on the air quality in the Bay Area. In the Oct 11 daytime plume, hourly $PM_{2.5}$ concentration in Napa and Vallejo exceeded 350 μg m$^{-3}$. $PM_{2.5}$ levels in other BAAQMD monitoring stations

(with locations shown in Fig. S3) reached peaks of above 100 μg m$^{-3}$ successively, roughly following the distances from the fires. Even the San Jose Jackson Street site (around 120 km downwind of the fires) recorded hourly averaged $PM_{2.5}$ of ~120





µg m$^{-3}$. The Oct 11 nighttime plume and the Oct 17 daytime plume were less dense in terms of PM$_{2.5}$. The temporal profiles of PM$_{2.5}$ at different measurement sites clearly show that they were all affected by the same plumes. In Oct 2017, when there was no influence from the wildfires, the PM$_{2.5}$ at these BAAQMD sites typically stayed below 15 µg m$^{-3}$. Therefore, when

the smoke came, the BBOA was the dominant component of particulate matter.

### 3.2  Chemical composition of particle-phase organic aerosols

The concentrations of different classes of compounds in each sample measured by the GC×GC are displayed in Fig. 3. In the three samples with highest total quantified mass (all from cluster 3 plumes), the compounds quantified can explain 15-20% of total OC by mass. However, in samples with minimal biomass burning influence only 5-10% of OC can be explained. We

define periods with total quantified OA above 4 µg m$^{-3}$ as plume periods (17 samples), total quantified OA < 0.8 µg m$^{-3}$ as background periods (30 samples), and samples with OA in the middle as BB influence periods (27 samples). The background periods were not totally unaffected by BBOA. The mean and standard deviation of concentrations for each group of compounds, the sum of all measured compounds (total quantified OA), and the total OC in the three kinds of periods are shown in Table 1. Sugars, dominated by levoglucosan, account for more than a third of total quantified OA in

plume periods. Based on the structures, most of these sugars (if underivatized) can fragment into C$_2$H$_4$O$_2^+$ (*m/z* 60) under EI (Fabbri et al., 2002). Terpenoids (especially resin acids) and nitrogen-containing compounds (dominated by nitrocatechols) were also specific to BBOA, while other families of compounds had other sources. For instance, the acids were enriched when smoke was affecting Berkeley, but their fractions in OA are lower than in background plumes because of dilution by the BB specific compounds.


In all samples, the fraction of PAHs remained below 0.3% of total quantified OA. The low PAH fraction in OA measured at UCB could be a result of both low PAH emission and photochemical loss. The emission, exposure, and health impacts of PAHs in biomass burning received a great deal of attention in previous studies (Shen et al., 2013; Sun et al., 2018; Tuet et al., 2019). However, recent work has shown that the toxicity of smoke aerosols is better correlated with total PM$_{2.5}$ or OC than with total PAHs (Bølling et al., 2012; Kim et al., 2018). Other groups of compounds, such as monocyclic aromatic

compounds including hydroquinone, catechol and cinnamaldehyde (Leanderson and Tagesson, 1990; Muthumalage et al., 2018), may also make substantial contributions to the toxicity of biomass burning smoke. The effect of compounds other than PAHs may be worthy of attention in future toxicological studies of BBOA. The maximum concentrations of the most abundant positively identified compounds and most abundant PAHs observed at Berkeley and their possible hazards are

listed in Table S1. Knowledge of the health impacts of inhaling compounds in this list are still lacking. For example, many nitroaromatic compounds were found to be mutagenic (Purohit and Basu, 2000). The nitro-compounds were found to be the main contributor to the mutagenicity of PM$_{2.5}$ in Northern Italy (Traversi et al., 2009). The sum of concentrations of (methyl-)nitrocatechols observed at Berkeley exceeded 1.2 µg m$^{-3}$. However, no toxicological research of these compounds was found in PubChem. These compounds could be useful candidates for future toxicological studies.





Figure 4 displays the volatility and $\overline{OS_c}$ distribution of speciated compounds in two relatively fresh samples (with 3-4 hours aging). The two samples have almost equal total quantified OA by mass. In the two samples, the volatility distribution and the $\overline{OS_c}$ distribution were almost identical. The volatility distribution obtained is similar to particle-phase primary BBOA reported by Hatch et al. (2018) in the way that most compounds reside in $10^0 < C^* < 10^2$ µg m$^{-3}$ bins. But the standard deviation of $\log_{10}C^*$ in our study is higher than that in Hatch et al. (2018), which could be related to fuel differences and

aging. Compounds in the $10^{-2}$ µg m$^{-3}$ < C* < $10^{-1}$ µg m$^{-3}$ bin were mainly aliphatics and tri-terpenoids, while 1 µg m$^{-3}$ to $10^2$ µg m$^{-3}$ bins consist of sugars, aromatic compounds, and other oxygenated compounds. The $\overline{OS_c}$ distributions measured at UCB differ remarkably from that inferred from thermodenuder + AMS measurements of primary BBOA from wood combustion (Donahue et al., 2012; Grieshop et al., 2009c). In that measurement, most of compounds had $\overline{OS_c}$ near -1.5. However, our measurement shows although there is a peak of OA with $\overline{OS_c}$ between -2 and -1.5 contributed mainly by

mono-carboxylic acids, sugars and oxygenated species cause a larger peak of $\overline{OS_c}$ between -0.5 and 0.5. The fragmentation probability is an important parameter for simulating SVOCs in fire plumes (Alvarado et al., 2015). The probability of an SVOC compound to fragment when reacting with OH can be estimated by $p = (O:C)^{0.25}$ (Donahue et al., 2013). Assuming $\overline{OS_c} \approx 3(O:C) - 2$, for compounds with $\overline{OS_c}$ between -0.5 and 0.5, the probability for them to fragment and form more volatile compounds are roughly 0.84 - 0.96. Stronger fragmentation could reduce net growth of mass of particle-phase

BBOA in aging processes.

### 3.3.1 Primary BBOA markers can indicate vegetation burned

Figure 5 shows the timelines of OC, commonly used BB marker compounds and the backward trajectory clusters throughout the campaign. Levoglucosan and mannosan, being the decomposition products of cellulose and hemicellulose, respectively, are emitted in the combustion of most plants (Jen et al., 2019; Nolte et al., 2001; Simoneit, 2002). Levoglucosan is the most

abundant BBOA species measured in this campaign. Its abundance reached around 20% of total quantified OA in the plume periods. Comparing Fig. 5d and 5e, when there was a peak of OC, there were usually peaks of levoglucosan as well. The levoglucosan to mannosan mass ratio can be used to differentiate hardwood and softwood (conifer) fires. Hardwood fires usually have emission ratios of levoglucosan/mannosan around 20, while for softwood fires this ratio is usually less than 5 (Cheng et al., 2013). The levoglucosan to mannosan ratio stayed above 20 in most samples, which confirms the dominance

of hardwood as fuel in the October 2017 Northern California wildfires as shown in Figure 1 (c) and (d).

Diterpenoids including resin acids are unique markers in biomass burning emissions for conifer combustion (Hays et al., 2002). Figure 5c shows the time series of dehydroabietic acid (DHAA), di-dehydroabietic acid (di-DHAA), abietic acid and retene. The concentration of the most abundant resin acid, dehydroabietic acid, reached over 0.8 µg m$^{-3}$ in a plume. These

conifer tracers mainly showed up in three plumes on Oct 11, 12 and 13. Figure 5a shows those plumes were associated with backward trajectories in cluster 2 and 3 which mainly transported smoke form the fires in Sonoma County, in which more



than 20% of the vegetation burned was conifer. In contrast, the BB plumes on October 10 and 17 were not accompanied by peaks of these conifer fire makers. That agrees with the fact that those plumes originated in the northeast (Fig. 5a), which mainly transported smoke from the Atlas Fire with little conifer combustion. Dimethopxyphenols and amyrins are markers

of hardwood. They were enriched in the plumes on Oct 10 and 17, as well as the plumes on Oct 11-13, which further confirms the ubiquity of hardwood fuels in all the fires affecting Berkeley. Hydroquinone and two other compounds were also shown to be good tracers for manzanita fires (Jen et al., 2018). They were present in most of the plumes, which is in line with the prevalence of manzanita in that region.

### 3.3.2 Traditional specific secondary BBOA markers

We focus on the behaviors of two groups of BB-specific SOA compounds here. 7-oxo-dehyroabietic acid (7-oxo-DHAA) was proposed to be an aging product of resin acids (Yan et al., 2008). However, other studies have suggested that DHAA can thermally degrade to 7-oxo-DHAA and finally retene in fires, and the differences in the abundance of these compounds can be attributed to the combustion temperature (Ramdahl, 1983; Simoneit et al., 1993; Standley and Simoneit, 1994). To figure out whether this process mainly occurs in the fires or in the atmosphere, a comparison between source and receptor profiles

of these compounds is needed. The ratios and concentration timelines of DHAA, 7-oxo-DHAA and retene are shown in Figure 6a and 6b. Both the 7-oxo-DHAA/DHAA ratio and the retene/DHAA ratio reached peak after the peak of DHAA. In addition, as shown in Figure S4, the 7-oxo-DHAA/DHAA ratio and the retene/DHAA ratio in the primary BBOA were far below the ratios detected in the ambient samples at Berkeley. Therefore, it is likely that the conversion to 7-oxo-DHAA (and retene) mainly happened through oxidation during transport in the atmosphere. We conclude that the 7-oxo-DHAA to

DHAA ratio provides a useful indicator for the formation of BB SOA.

Another group of compounds often used as tracers for BBSOA in source apportionment are the nitro-aromatic compounds (Watson et al., 2016). When catechol or methyl-catechols react with OH, $NO_3$ or HONO with the presence of $NO_2$, 4-nitrocatechol or methyl-nitrocatechols will be produced (Bertrand et al., 2018; Finewax et al., 2018; Iinuma et al., 2010;

Vidović et al., 2018). The low vapor pressures of these compounds cause them to be mainly in the particle phase, and therefore are expected to be useful BB SOA markers (Finewax et al., 2018). Figure 6d shows the concentration time series of the nitrocatechols. The sum of the nitrocatechols reached 1.37 μg m$^{-3}$ in the early morning on Oct 13, which is 9.1% of total quantified OA. The nitrocatechol/OC ratios in daytime plumes (Oct 11, Oct 12 & 13 afternoons) were lower than in the nighttime/early morning plumes (Fig. 6c). Since the plumes came from the same fires, the diel difference was either caused

by the differences in day/night combustion processes or oxidation chemistry. The concentration of catechol and 4-nitrocatechol are shown in Fig. S5. In the daytime plumes, there were more catechol relative to 4-nitrocatechol than in the nighttime/early morning plumes. Finewax et al. (2018) has shown that the molar yield of 4-nitrocatehcol when catechol reacts with OH and $NO_3$ are 0.3 ± 0.03 and 0.91 ± 0.06, respectively. The difference in oxidation mechanism is thus a more plausible explanation to the diel difference of nitrocatechols/OC ratio.





Qualitatively, the nitrocatechols are good markers for BB SOA. However, assuming [OH] = $2 \times 10^6$ molecules cm$^{-3}$, and
[NO$_3$] = $5 \times 10^8$ molecules cm$^{-3}$, the lifetimes of catechol against OH or NO$_3$ oxidation are only 1.4 h and 20 s, respectively
(Finewax et al., 2018). As shown in Fig. 6b and 6d, the timelines of nitrocatechols are very similar to the primary BBOA
tracers like dehydroabietic acid, especially in the nighttime plumes. Nitrocatechols and dehydroabietic acid were categorized
into the same timeline factor in all 4-factor to 8-factor solutions in the HCA analysis (see Sections 2.6 and 3.4). Although

these smoke plumes traveled approximately 3-4 hours from the fires to Berkeley, it was long enough for the formation of
substantial amounts of nitrocatchols. In addition, 4-nitrocatechol was also detected in fresh BBOA in the Fire Lab study (Jen
et al., 2019). This issue could be a challenge of using nitrocatechols as BB SOA markers in timeline-based source
apportionment analyses.

### 3.4 Clustering of compounds by HCA

To reduce the complexity of interpreting the time series of each of the 572 compounds, we simplified them into 7 factors
based on the similarity in temporal behavior. Figure 7a displays the dendrogram of the factors. The predicted volatility and
calculated/predicted $\overline{OS_c}$ of each compound are displayed in Fig. 7b, colored by the factor the compound belongs to. Factor 1
($N = 85$) compounds include levoglucosan, mannosan, vanillic acid, 4-hydroxybenzoic acid and pentacosanoic acid, etc.
These compounds are universally emitted in burning most kinds of biomass fuels. The grass fire tracer, $p$-Coumaryl alcohol

(Nolte et al., 2001; Oros et al., 2006; Simoneit et al., 1993), is also in this factor. It indicates the presence of grass as
understory fuel in all the fires. This factor contributes around 1/3 of the total quantified mass (Fig. 8). Factor 2 ($N = 78$) is
also dominated by primary BBOA compounds. Most aliphatic acids, alcohols, and alkenes above C$_{20}$ (including $n$-
nonacosan-10-ol) are in this factor. The most abundant alkane measured, the $n$-nonacosane, is also in this factor. These
compounds are likely from the plant wax sources (Medeiros and Simoneit, 2008; Simoneit, 2002). Like Factor 1, compounds

in the second factor were present in almost all BB plumes. Factor 3 ($N = 100$) is also a primary BBOA factor. But it is hard
to associated with it with any specific fuels based on the chemical composition, and it is therefore called the "primary BBOA
unknown" factor. Compounds in this factor were more abundant in the Oct 11 daytime plume than the Oct 12 and 13
plumes. This factor contains both hardwood tracers like syringic acid, as well as galactosan and pinitol, which are more
abundant in conifer combustion emissions (Medeiros and Simoneit, 2008; Munchak et al., 2011). Other sugar alcohols, such

as $myo$-inositol, ononitol, erythritol and xylitol are in this factor. These compounds are more abundant in green (moist)
vegetation fires (Medeiros and Simoneit, 2008; Schmidl et al., 2008). The most abundant sugar alcohol $myo$-inositol is
present in both plants and animals as well (Loewus and Murthy, 2000; Medeiros and Simoneit, 2008). Its concentration
reached 0.28 μg m$^{-3}$ in the Oct 11 plume originated from the Sonoma County.

Factor 4 ($N = 35$) is the most volatile factor, with more than half the compounds having C* above $10^2$ μg m$^{-3}$. This factor
contains urban OA, such as nicotine, delta-9-tetrahydrocannabinol, glycerol, and terephthalic acid. However, several
compounds emitted in hardwood fires, such as pyrogallol, syringol and sinapyl alcohol are also in this factor. Their


concentrations in the Oct 17 plumes were comparable to or even higher than in the Oct 11-13 plumes. Since the Oct 17 plume mainly came from the Atlas Fire, the high abundances of these tracers were expected. However, it is unclear why

other hardwood BBOA markers like syringaldehyde and sinapaldehyde did not follow this trend. The concentrations of syringol and pyrogallol were found to increase moderately in aging experiments due to partitioning or chemical formation (Bertrand et al., 2018; Fortenberry et al., 2018). Since the Oct 17 plume were more aged than the Oct 11-13 ones, it is possible that the two compounds were formed during transport. Factor 5 ($N$ = 124) is a BBOA factor. Its concentrations exceeded Factor 1's in the densest plumes on Oct 11-13. This factor represents BBOA from the Sonoma County fires

according to the backward trajectory cluster. It consists of many hardwood tracers (such as sinapaldehyde, lupeol, α-amyrin and syringaldehyde) and softwood tracers (such as DHAA, pimaric acid, sandaracopimaric acid, abietic acid and retene), as well as fast-forming SOA such as nitrocatechols. The co-occurrences of both hardwood and softwood tracers indicates these fuels were burned simultaneously in the fires in Sonoma County.

Factor 6 ($N$ =100) is the daytime BBSOA factor. This factor has a moderate correlation with levoglucosan ($r$ = 0.68). The dendrogram shows this factor has large distance (small correlation) from any other factor (Fig. 7a). Figure 9 displays the concentration of this factor versus time. The major peaks of this factor either happened in the afternoon or in the plume that arrived in Berkeley at night but previously experienced daytime aging. Higher daytime concentrations of these compounds indicate stronger aging processes of BBOA in daytime. The Oct 11 afternoon plume was relatively fresh, with ~3 hours

daytime aging (in backward trajectory cluster 2). The Oct 12 & 13 afternoon plumes were more aged (in backward trajectory cluster 3). Although the OC level in the Oct 11 plume was the highest, the concentration of Factor 6 was higher in Oct 12 & 13 afternoon plumes. This factor accounted for 9-14% of total quantified OA in those plumes. More than 90% of the mass of this factor resides in C* volatility bins between 1 and $10^3$ μg m$^{-3}$, which indicates the semivolatile nature of this factor. Positively identified BBSOA compounds include multifunctional aliphatic acids and oxygenated aromatic compounds (Table

2). All the multifunctional aliphatic acids have less than 10 carbon atoms. Although they were abundant in aged biomass burning plumes, many of them are not specific tracers of BB SOA. For instance, malic acid and tartaric acid were found in aged wood smoke in oxidation experiments (Hartikainen et al., 2020). However, they can be produced in the oxidation of 1,3-butadinene and isoprene (Claeys et al., 2004; Jaoui et al., 2014). Malic acid can also be produced by the hydroxylation of succinic acid, an oxidation product of long-chain unsaturated fatty acids (Kawamura et al., 1996; Kawamura and Ikushima,

1993). Pinic acid and 3-methyl-1,2,3-butanetricarboxylic acid (MBTCA), commonly used as biogenic (monoterpene) SOA tracers (Jenkin et al., 2000; Szmigielski et al., 2007; Zhang et al., 2018), are also in this factor. Monoterpenes and oxy-monoterpenes can account for more than 5% of total non-methane organic gas emission in certain conifer fires (Hatch et al., 2019). The biogenic SOA could be oxidation products of the terpenes emitted in fires. It is also possible that biomass burning emissions enhanced the formation of biogenic SOA. As for aromatic compounds listed in Table 2, phthalic acid, 4,7-

dimethyl-1,3-isobenzofurandione and 1,3-dyhdroxynaphthalene could be PAH oxidation products (Wang et al., 2007). Protocatechuic acid and gallic acid are likely more specific to biomass burning. Protocatechuic acid is a product of coniferyl





alcohol and coniferyl aldehyde ozonolysis, and the vanillic acid + $NO_3$ reaction (Liu et al., 2012; Net et al., 2010, 2011). Protocatechuic acid and gallic acids were also found to be the Fenton-like oxidation products of biomass burning-related small aromatic acids in the atmospheric aqueous phase (Santos et al., 2016a, 2016b). These compounds could be monitored

in future field and lab studies to verify whether they are suitable BB SOA tracers.

Factor 7 ($N = 51$) is an urban OA factor. Palmitic acid, stearic acid, benzophenone, 6,10,14-trimethyl-2-pentadecanone are all in this factor. The concentration of this factor was more constant over time than the others. It did not sharply increase in the fire plumes. EC (not included in Fig. 8) was also grouped into this factor, which indicates that EC measured at the UCB

campus was not dominated by EC from fires.

**3.5 Effect of aging on the mass of fire induced PM$_{2.5}$ and OC**

To investigate whether PM$_{2.5}$ mass increased or decreased during transport and aging, the ΔPM$_{2.5}$/ΔCO ratio at multiple BAAQMD sites affected by the Oct 11, 12 and 17 plumes are compared in Figure 10. The levels of PM$_{2.5}$ at the sites considered here reached peak 0-5 hours after the Napa site. The ΔPM$_{2.5}$/ΔCO ratios in each fire plume intercepted at

different sites have a narrow distribution, epecially in the Oct 11 noon time plume. PM$_{2.5}$ were diluted by 2 to 3 times in 2 hours. In this process, the evaporation induced by dilution must have approximately balanced SOA formation. Minimal increase of particle OC mass in aging were observed at UCB campus as well. Acetonitrile is a stable compound that is commonly used as a biomass burning tracer (Gilman et al., 2015; Holzinger et al., 2005). Figure 11 shows that particle OC is linearly correlated with acetonitrile. The relationship was not affected by the fraction of the BB SOA factor in total

quantified OA or day/night difference. The strong correlation and other indicators suggest that although substantial chemical transformation happened, there must have been near balance between evaporation and secondary OC formation in terms of the particle OC budget. The evaporation (and fragmentation type oxidation) of primary BBOA could reduce the net increase of PM$_{2.5}$ and OC mass. Also, since many BB SOA compounds are also semivolatile, their evaporation and further oxidation could also reduce the net increase of particle mass. Since aged BBOA has slightly higher O/C ratio, it can be inferred that

aging still resulted in a small increase of BBOA mass in aging. However, this increase was much smaller than the results reported by Vakkari et al. (2018), in which PM$_1$ mass more than doubled in only three hours of daytime aging.

**4. Conclusions**

The chemical composition of organic aerosol during the October 2017 Northern California wildfires was characterized in detail, tracking nearly 600 chemicals at 3-4-hour time resolution. We demonstrated that using easily obtained parameters

from GC × GC measurements, the volatility and $\overline{OS_c}$ of compounds can be satisfactorily predicted. The OA consisted of compounds spanning 10 orders of magnitude in volatility. The BBOA had high $\overline{OS_c}$, and possibly high chance to fragment. We found that the time series of primary BBOA tracers at the receptor site can be used to trace back the fuels burned, and

the timelines of BB SOA markers can indicate the transformation processes. Through hierarchical clustering analysis, we traced back the sources of the OA measured at Berkeley and discovered a unique daytime BB SOA factor. Compounds in
that factor are highly oxygenated but are still semi-volatile. Using the $PM_{2.5}$ and CO measured by the BAAMD network, we found the growth of particle mass during aging was small. Similar analysis could be used to study other fires. Relatively fresh and aged samples had similar OC/acetonitrile ratio, which indicates the evaporation of particle organic compounds and the condensation of gas phase organic compounds were balanced in terms of carbon.

The thermal desorption GC × GC measurements remarkably improved our ability to separate and quantify the chemicals residents in the San Francisco Bay Area were exposed to. We hope the resulting database can contribute to more accurate exposure assessments from wildfire smoke in general. Also, with better speciation of compounds in wildfires, more targeted toxicological studies could be carried out to elucidate the health impacts of BBOA.

## 5. Data availability

The concentration timelines of each compound, and the information of each compound will be deposited on https://nature.berkeley.edu/ahg/resources/. The data used in this research is also available from authors upon request.

## 6. Author contribution

Y.L., C.N.J and P.K.W carried out measurements at UC Berkeley. Y.L., C.N.J and R.J.W. performed experiments at the Advanced Light Source. Y.L., C.N.J, R.J.W. and A.H.G. analyzed data. Y.L. and A.H.G wrote the manuscript with all
authors contributing comments.

## 7. Acknowledgements

This work was sponsored by an NSF RAPID grant (award number: 1810641). The authors thank BAAQMD for providing air quality and meteorological measurement data in the Ambient Air Monitoring Network. We acknowledge the use of data and imagery from LANCE FIRMS operated by NASA's Earth Science Data and Information System (ESDIS) with funding
provided by NASA Headquarters. The authors thank Dr. Jeremy Nowak, Bruce Rude and Dr. Kevin Wilson for their assistance during the beamline campaign. This research used resources of the Advanced Light Source, which is a DOE Office of Science User Facility under contract no. DE-AC02-05CH11231. The authors thank Dr. Deep Sengupta at UC Berkeley for helpful discussions.



## 8. Competing interests

The authors declare that they have no conflict of interest.



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



**Table 1:** Concentration of each chemical family, total quantified OA by GC × GC, and OC [µg m$^{-3}$] of the three regimes.

| Compound family | BB plume periods | | BB influence periods | | Background periods | |
|---|---|---|---|---|---|---|
| | **Mean** | **St. Dev.** | **Mean** | **St. Dev.** | **Mean** | **St. Dev.** |
| Acid | 1.25 | 0.39 | 0.43 | 0.18 | 0.18 | 0.11 |
| Alcohol | 0.20 | 0.08 | 0.05 | 0.03 | 0.02 | 0.01 |
| Alkane & alkene | 0.16 | 0.05 | 0.08 | 0.04 | 0.03 | 0.03 |
| Aromatic | 1.01 | 0.46 | 0.19 | 0.13 | 0.08 | 0.03 |
| Nitrogen-containing | 0.56 | 0.48 | 0.10 | 0.09 | 0.03 | 0.03 |
| Other oxygenated | 1.12 | 0.32 | 0.34 | 0.18 | 0.14 | 0.05 |
| PAH | 0.03 | 0.03 | <0.01 | <0.01 | <0.01 | <0.01 |
| Sugar | 3.39 | 0.40 | 0.46 | 0.28 | 0.05 | 0.06 |
| Terpenoid | 0.40 | 0.39 | 0.03 | 0.04 | <0.01 | <0.01 |
| Unknown | 0.23 | 0.06 | 0.05 | 0.02 | 0.02 | 0.01 |
| Total quantified OA [µg m$^{-3}$] | 8.28 | 3.82 | 1.75 | 0.71 | 0.55 | 0.16 |
| OC [µg m$^{-3}$] | 42.2 | 10.2 | 17.8 | 6.4 | 7.7 | 3.4 |




**Table 2** Concentrations of selected positively identified compounds in factor 6 in the plumes in Oct 11, 12 and 13 afternoons.

| Compound | Formula | Concentration [ng m$^{-3}$] | | |
|---|---|---|---|---|
| | | 11-Oct | 12-Oct | 13-Oct |
| **Multifunctional aliphatic acids** | | | | |
| 3,4-dihydroxybutanoic acid | $C_4H_8O_4$ | 13.3 | 13.9 | 20.3 |
| citramalic acid | $C_5H_8O_5$ | 0.0 | 8.4 | 11.5 |
| pimelic acid | $C_7H_{12}O_4$ | 26.8 | 26.0 | 26.0 |
| 2,2-bis(hydroxymethyl)propionic acid | $C_5H_{10}O_4$ | 17.5 | 32.8 | 16.9 |
| malic acid | $C_4H_6O_5$ | 32.3 | 32.8 | 50.5 |
| suberic acid | $C_8H_{14}O_4$ | 0.0 | 11.4 | 11.9 |
| threonic acid | $C_4H_8O_5$ | 14.9 | 10.3 | 9.9 |
| 2,3,4-trihydroxybutyric acid | $C_4H_8O_5$ | 7.3 | 4.9 | 4.5 |
| 2,3-dimethylsuccinic acid | $C_6H_{10}O_4$ | 5.2 | 5.3 | 5.8 |
| α-ketoglutaric acid | $C_5H_6O_5$ | 0.0 | 7.4 | 3.7 |
| 2-pentenedioic acid | $C_5H_6O_4$ | 5.7 | 2.8 | 3.1 |
| maleic acid | $C_4H_4O_4$ | 0.0 | 6.5 | 6.3 |
| pinic acid | $C_9H_{14}O_4$ | 12.7 | 8.2 | 10.3 |
| 2,3-dihydroxy-4-oxo pentanoic acid | $C_5H_8O_5$ | 2.2 | 5.9 | 9.3 |
| 3-methyl-1,2,3-butanetricarboxylic acid (MBTCA) | $C_8H_{12}O_6$ | 2.6 | 0.0 | 2.4 |
| **Oxygenated aromatic compounds** | | | | |
| 4,7-dimethyl-1,3-isobenzofurandione | $C_{10}H_8O_3$ | 0.8 | 1.3 | 2.0 |
| 1-phenyl-1-penten-3-one | $C_{11}H_{12}O$ | 2.4 | 6.4 | 9.1 |
| 1,3-dihydroxynaphthalene | $C_{10}H_8O_2$ | 6.3 | 3.3 | 3.3 |
| protocatechuic acid | $C_7H_6O_4$ | 43.5 | 25.3 | 35.2 |
| phthalic acid | $C_8H_6O_4$ | 27.7 | 45.9 | 68.5 |
| gallic acid | $C_7H_6O_5$ | 2.2 | 2.6 | 2.6 |





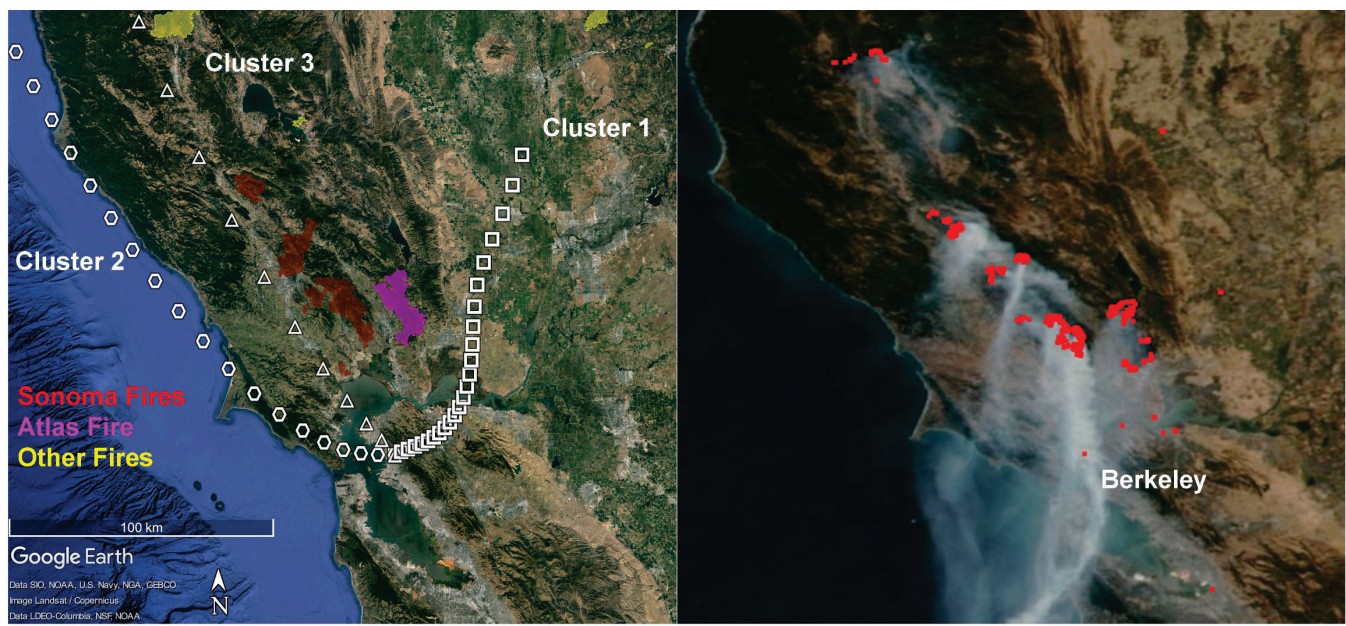

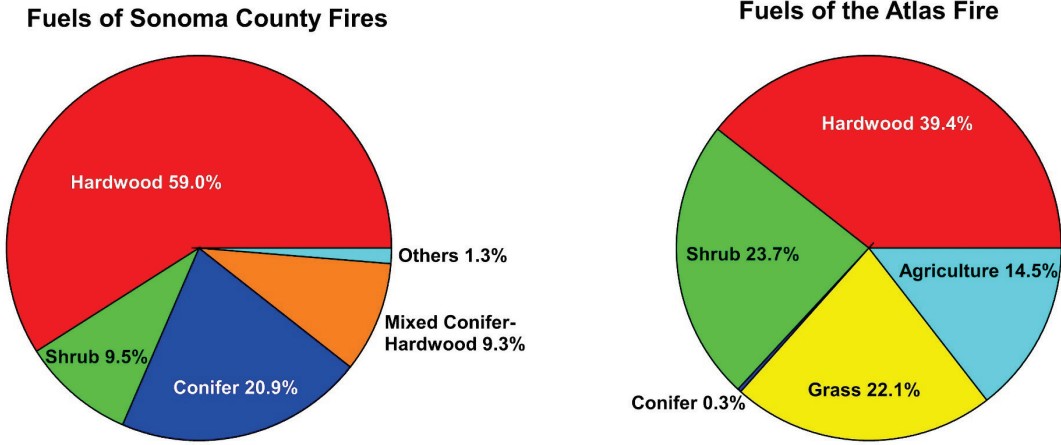

**Figure 1: (a)** Perimeters of fires, and the mean backward trajectory of each cluster with one label per hour **(b)** satellite image of this area on October 12, 2017, with fire points detected by VIIRS labelled by red dots **(c)** fuel composition of the fires in the Sonoma County **(d)** fuel composition of the Atlas Fire in Napa County.





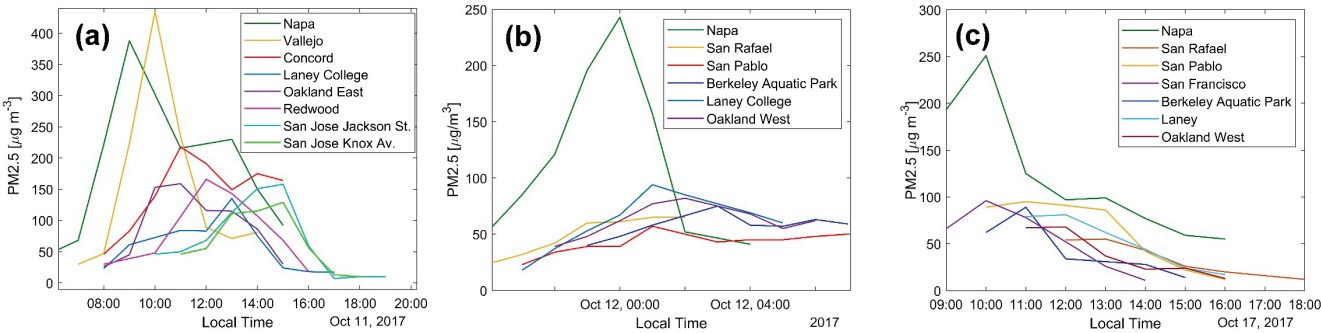

**Figure 2:** Hourly PM$_{2.5}$ concentration measured at BAAQMD sites affected by the fires plumes on **(a)** October 11, 2017, **(b)** October 12, 2017 and **(c)** October 17, 2017

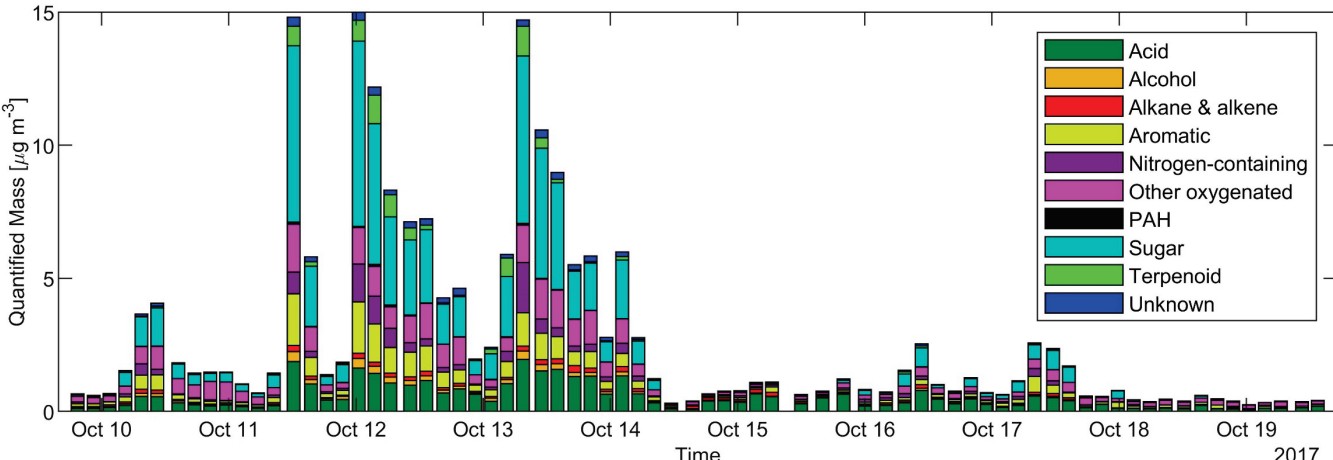

**Figure 3:** Concentrations of speciated compounds classified by chemical family in each sample. Each stacked bar shows the sum of concentrations of compounds in each class in a 3 h or 4-h sample.



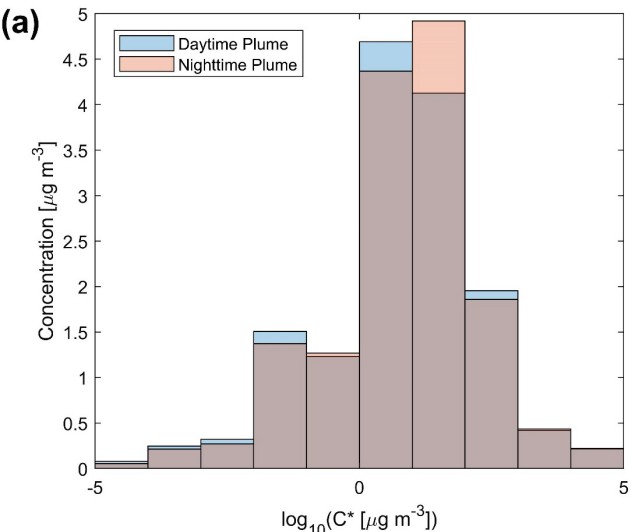

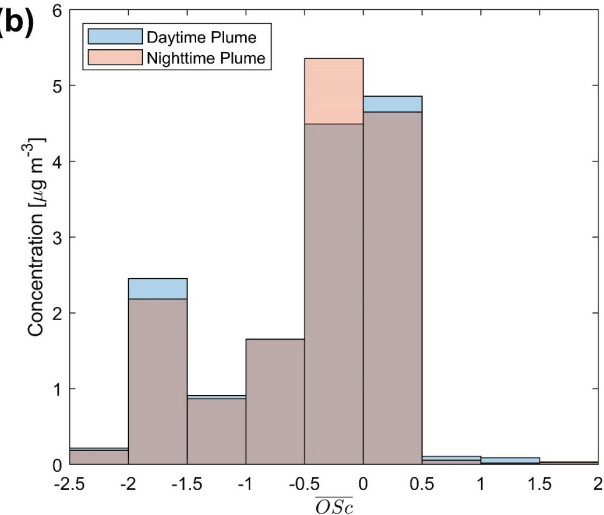

**Figure 4:** Mass distibution of speciated compounds in different **(a)** effective saturation concentration and **(b)** average carbon oxidation state bins. The daytime plume is from a sample in the Oct 11 plume described in Figure 2a. Nighttime sample is from a sample in the Oct 11-12 plume in Figure 2b. The gray parts of the bar show the overlaps of the daytime plume and the nighttime plume samples.





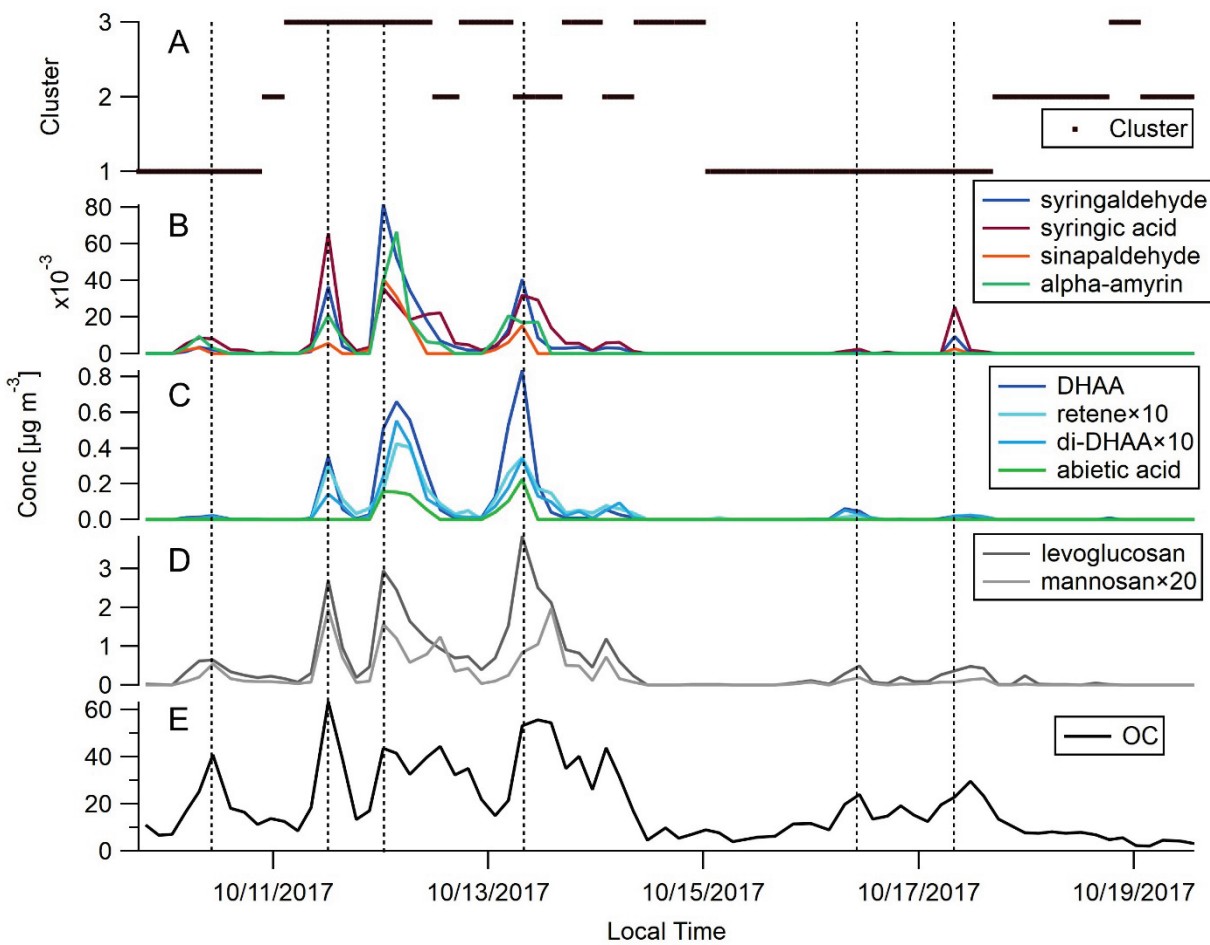

**Figure 5:** Time series of **(a)** backward trajectory cluster **(b)** hardwood primary BBOA markers and **(c)** softwood primary BBOA markers **(d)** general primary BBOA markers and **(e)** OC






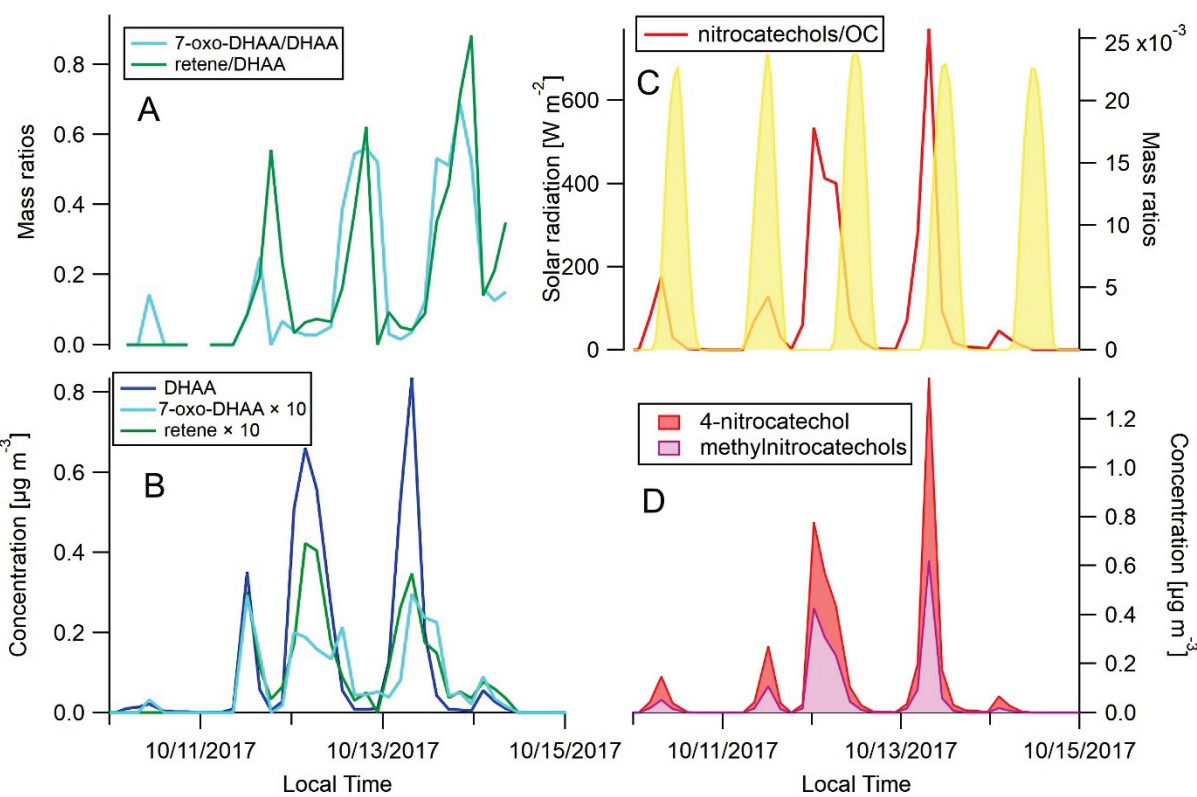

**Figure 6:** Time series of **(a)** mass ratios of 7-oxo-DHAA/DHAA and retene/DHAA **(b)** concentrations of DHAA, 7-oxo-DHAA and retene **(c)** mass ratios of the sum of nitrocatechols/OC and solar radiation **(d)** concentrations of 4-nitrocatechols and methylnitrocatechols. Solar radiation in (c) was measured at Bethel Island, a measurement site in the middle of Napa and Berkeley.






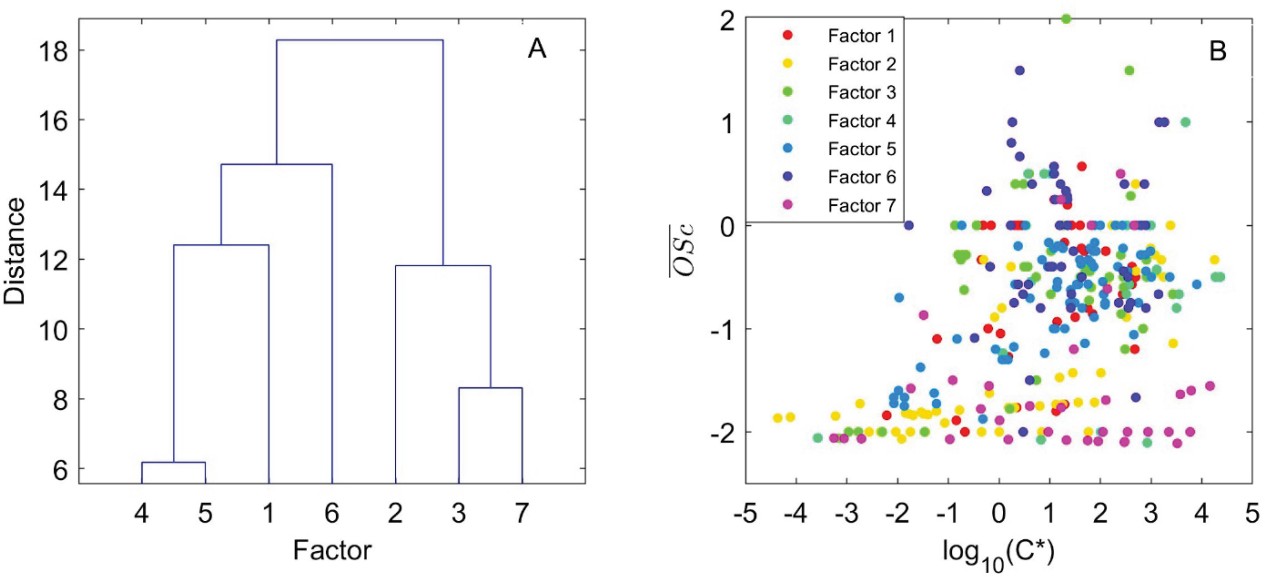

**Figure 7: (a)** Hierachical clustering relationship **(b)** Effective saturation concentration vs. $\overline{OSc}$ of compounds colored by factor number.

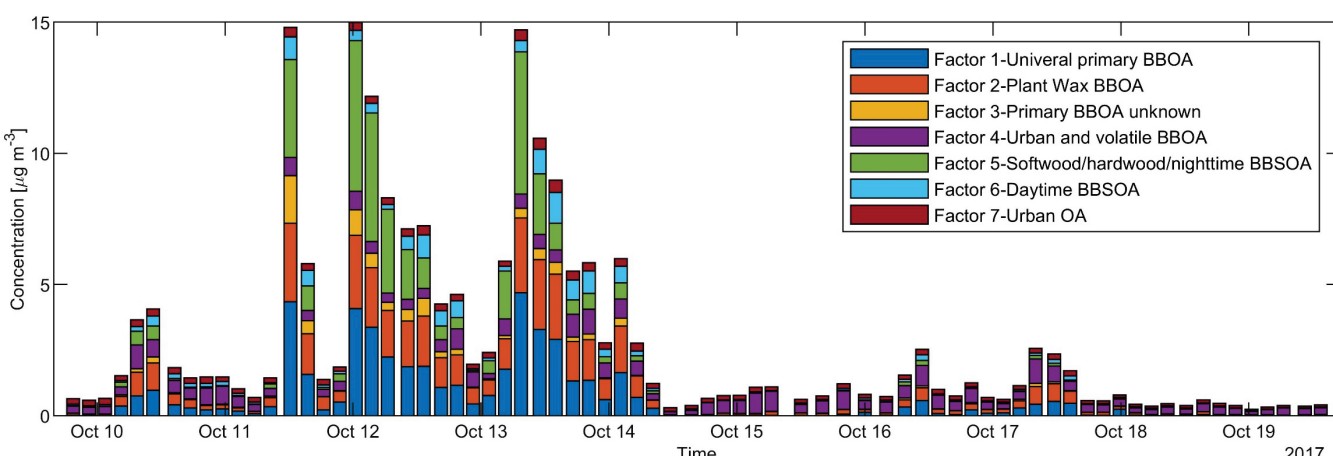

**Figure 8:** Sum of concentrations of GC × GC speciated and quantified compounds in different factors in each sample.




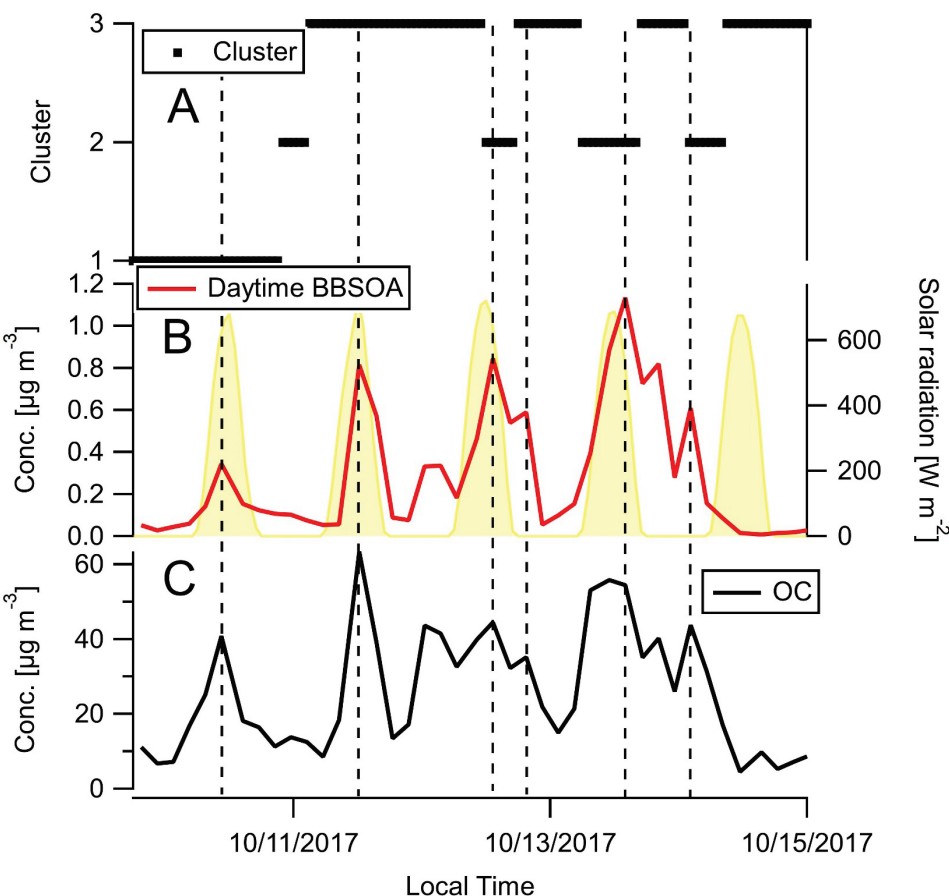

**Figure 9:** Time series of **(a)** backward trajectory cluster **(b)** concentration of compounds in the daytime BBSOA factor and **(c)** OC. Solar radiation in (b) was measured at Bethel Island, a measurement site in the middle of Napa and Berkeley.



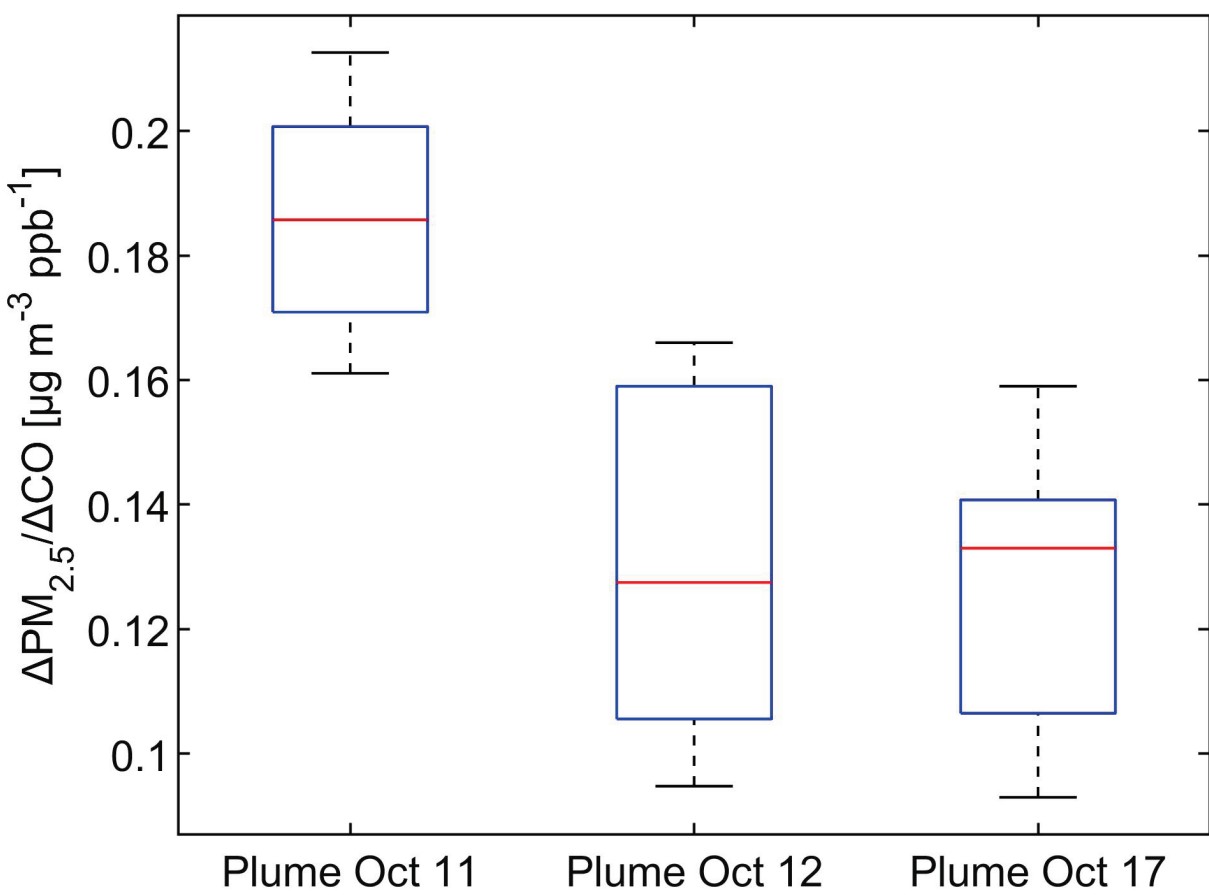


**Figure 10:** Boxplot of the Δ PM$_{2.5}$ ΔCO in µg m$^{-3}$ ppb$^{-1}$ at BAAQMD sites in the three plumes described in Fig. 2. ΔPM2.5/ΔCO were obtained by linearly fit PM$_{2.5}$ versus CO during plumes (using $R^2 = 0.9$ as the threshold). The top and bottom edges of the box indicate the 75$^{th}$ and 25$^{th}$ percentiles, respectively. The red central mark is the median. Whiskers extend to 1.5×interquartile ranges out of the boxes. There was no outlier in the data.

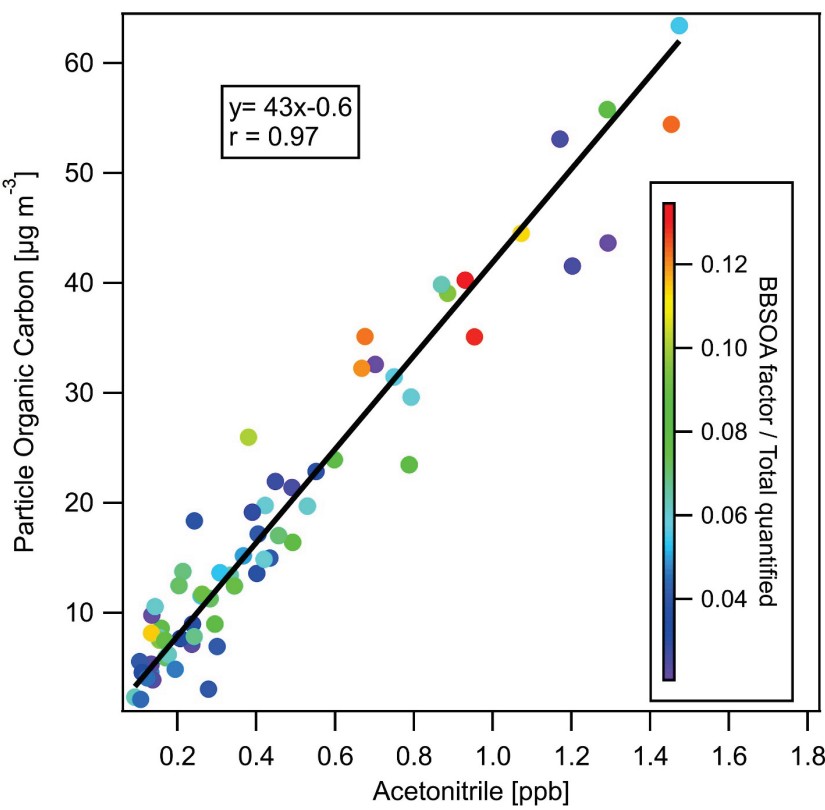


**Figure 11:** Particle-phase OC vs acetonitrile (measured by PTR-ToF-MS), colored by the ratio of BBSOA factor to total GC × GC speciated and quantified OA. Linear regression line (using orthogonal distance regression) is also displayed.