# Peer review of "Chemical Composition of PM2.5 in October 2017 Northern California Wildfire Plumes"

_Atmospheric Chemistry and Physics, 2020_

## Referee Comment (RC1) · Anonymous Referee #1 · 29 Oct 2020

**Summary/recommendations:**

Liang et al. have provided a strong, thorough study of aged biomass burning emissions from California wildfires in 2017. There are several interesting observations within the paper based on many lines of evidence, including mean $OS_C$ distributions, volatility estimates, and fuel markers from different fire types. The research done for this study provides a nice framework for future BB studies. I recommend the paper be published after the following minor comments. As one point to be noted, there were a number of figures that could be slightly modified to make the paper more accessible and digestible, and I have provided specific comments below.

**General comments:**

Line 122: briefly explain what the 3-cluster solution is, and link to Figure 1a & sect 3.1.

Lines 246-251: Smoke age is estimated for cluster 3; can it be estimated and provided for clusters 1 and 2?

I would note where the subscript '74' comes from in Eq 2 when introducing the equation, I had to search for that (explained several sections earlier).

Sect 3.4: Do the authors have a hypothesis (or hypotheses) on why Factor 6 is the least correlated to any of the other factors?

I recommend briefly explaining what the dendrogram means in Fig 7a.

Sect 3.5: it may be worth pointing out that rapid chemistry and SOA formation may occur before the time of the first set of BAAQMD measurements. Could the authors note the estimated physical distance of the monitors closest to the fires, and provide the range of physical distances in monitors?

**Figures/tables:**

All of the time series figures could benefit and become much more digestible by clear, consistent markings of different periods (plume, background, etc). See below for more specifics:

Figure 3 & 8: consider marking the different periods discussed in sect 3.2 (plume, background, BB influenced) on the figure to increase its usefulness. (I personally find time series figures that have the background of the plot shaded different colors to indicate different periods to be particularly useful. For example, see figures 1-2 of Mattila et al., 2020)

Mattila, J. M., Arata, C., Wang, C., Katz, E. F., Abeleira, A., Zhou, Y., Zhou, S., Goldstein, A. H., Abbatt, J. P. D., DeCarlo, P. F. and Farmer, D. K.: Dark Chemistry during Bleach Cleaning Enhances Oxidation of Organics and Secondary Organic Aerosol Production Indoors, Environ. Sci. Technol. Lett., 0–6, doi:10.1021/acs.estlett.0c00573, 2020.

Figure 5 & 9: what are the vertical dashed lines? Are these marking specific plume periods? Please specify in the figure caption. Again, marking the different periods can make these types of figures more easy to digest. (See comment for Figure 3 & 8)

Figure 7b: the colors between 3 and 4 and between 1 and 7 are difficult to distinguish. I recommend making all of the colors contrast more here; could also consider using different shapes to help with the contrast.

**Technical comments:**
There are a number of and minor grammatical errors (primarily in the form of word omissions; e.g. lines 32, 43-44, 223) throughout the paper.

Line 257: 'The Oct 11 nighttime plume': this is referred to the Oct 12 plume in Fig 2. Please make sure how each plume is referred to (Oct 11, Oct 12, etc) is consistent throughout the paper.

Line 433-434: "The levels of PM2.5 at the sites considered here reached peak 0-5 hours after the Napa site." This sentence seems to be missing a word, such as "reached *their* peak"

---

## Referee Comment (RC2) · Anonymous Referee #2 · 23 Jan 2021

First, I apologize for the tardiness of my review! The paper is well written and informa-tion packed. The investigators effectively captured samples from several unique and interesting natural fire events with various levels of aging. This work is very valuable, an excellent addition to the existing literature, and an important development given the increasing importance of fire emissions in recent years. The paper builds upon exist-ing laboratory studies and does a fine job of addressing the difficulties and realities of real-world sampling.

This paper is not within my area of expertise (and it's been 20 years since I took organic chemistry) so the following comments may be irrelevant, but that said, I had a difficult time following some of the discussions and conclusions. The paper presents a wide range of topics and analyses. Little background is provided to bring the reader up-to-

speed on the various analyses techniques or chemical mechanisms. I wonder if the paper should be split into two papers so more thorough discussions can be provided.

There is no mention of data quality or uncertainties in the paper. These are very low concentrations and especially when ratios are presented, I question the reliability of the values.

I have a few specific comments on the text.

Line 33: Confusing sentence, suggest eliminating "primary and secondary" to simplify

Line 135: Why does the temperature only go up to 320C?

Lines 238-245: A comment on the accuracy of the forest inventories might be appropriate. I laughed at the 0.3% because these estimates have so much uncertainty.

Line 270-271: This sentence seems out of place. I don't understand how it fits into the current paragraph. What is EI?

Lines 279-288: This seems like introductory material since it is primarily on previous studies.

Lines 288-289: This seems out of place, like a discussion or conclusion point.

Line 326-327: Suggest rewrite of this sentence, as written it sounds like this work showed that hydroquinone and two other compounds were shown to be good tracers, but that's not the case.

Line 336: Suggest replacing "reached peak" with "peaked" for clarity

Line 339-340: This is a strong statement for what looks like a weak pattern in the figure. Suggest softening to something like "our observations were consistent with the hypothesis that 7-oxo-DHAA to DHAA ratio is a useful indicator..."

Line 354: Suggest adding a sentence addressing the diel concentration differences expected for OH and NO3 to complete this argument.

Line 450: "OSc of compounds can be satisfactorily predicted" This statement was not made in the previous sections, and I'm not clear on how you know it is "satisfactory".

Line 734: Instead of "label", I suggest using "symbol" in the caption

Figure 2: It is hard to distinguish the colors in these figures.

Figure 3: It would be easier to digest if the legend was in the same order as the stacked bars.

[Figure]

---

## Author Comment (AC1) · 26 Feb 2021

**Response to Reviewer 1**

We thank the reviewer for the insightful comments and suggestions, which helped to improve the quality of this article. We have addressed the reviewer's concern by making the following modifications to the paper.

**General comments:**

*Line 122: briefly explain what the 3-cluster solution is, and link to Figure 1a & sect 3.1.*

We changed this sentence to:

"We classified the air masses that arrived in Berkeley during the fire period into 3 clusters based on their origins, as shown in Figure 1a, and discussed in Section 3.1."

*Lines 246-251: Smoke age is estimated for cluster 3; can it be estimated and provided for clusters 1 and 2?*

We added the estimates into this paragraph, now it reads:

"The mean backward trajectory of each cluster is also shown on Fig. 1a. Plumes in cluster 1 arrived from the northeast at relatively low speed. **Their plume ages were quite variable, ranging from around 5 to 12 hours.** They mainly picked up smoke from the Atlas Fire. Plumes in cluster 2 originated from the west coast. They picked up smoke from the wildfires (mainly Sonoma County fires) and then transported it to the Bay Area. **The plume ages were estimated to be between 6 to 10 hours.** Smoke in cluster 2 is expected to be more aged than smoke in cluster 3. Plumes in cluster 3 traveled 3-5 hours from the fires to the UCB campus, as estimated from HYSPLIT. They mainly transported smoke from the Sonoma County Fires to the Bay Area."

*I would note where the subscript '74' comes from in Eq 2 when introducing the equation, I had to search for that (explained several sections earlier).*

We replace the 74 with *n*, and mention *n* = 74 here. Now it reads:

"The concentration timelines were first normalized to prevent all the high (or low) concentration compounds getting clustered into the same factor. Then the Euclidean distance between each pair of normalized timelines (e.g. compound concentration vectors u and v) is calculated by:

$$d_{uv} = \sqrt{(u_1 - v_1)^2 + (u_2 - v_2)^2 + ... + (u_n - v_n)^2} \qquad (2)$$

where $u_i$ and $v_i$ are the normalized concentrations of compound u and v at time step i, ***n* is the number of data points (*n* = 74 in this case)** respectively."

*I recommend briefly explaining what the dendrogram means in Fig 7a.*

As suggested, we add this short description to it, now it reads:

Figure 7a displays the dendrogram of the factors. **The dendrogram shows the hierarchical relationship between factors. The lower the distance is, the more correlated two factors are.**

**For instance, Factor 4 and Factor 5 have high correlation, but they are not well correlated with Factor 6.**

*Sect 3.5: it may be worth pointing out that rapid chemistry and SOA formation may occur before the time of the first set of BAAQMD measurements. Could the authors note the estimated physical distance of the monitors closest to the fires, and provide the range of physical distances in monitors?*

We added this after the first sentence of this paragraph:

"The Napa measurement station is around 5 km and 10 km from the perimeters of the Atlas Fire and the Nuns Fire, respectively. Therefore, the plumes captured at the Napa station already had ages of at least 10-20 minutes and likely longer. In this period, rapid chemistry and SOA formation might have taken place. The Vallejo measurement station is 24 km from the Napa station. Measurement stations in Berkeley, San Francisco and Oakland are 50-70 km away from the Napa station. The farthest measurement stations in San Jose are more than 110 km away from the Napa station."

We also added a scale to the map of the air quality measurement stations in Figure S3 in the Supplement.

**Figures/tables:**

Figure 3 & 8: consider marking the different periods discussed in sect 3.2 (plume, background, BB influenced) on the figure to increase its usefulness. (I personally find time series figures that have the background of the plot shaded different colors to indicate different periods to be particularly useful. For example, see figures 1-2 of Mattila et al., 2020)

We thank the reviewer for suggesting this. There are already multiple colors on this plot. We considered this suggestion, but decided that adding shading would make the plot too crowded. Unlike Mattila et al. (2020), we classified the plume/BB influenced/background periods only based on the total quantified OA of each sample, which has already been shown by the height of each bar. We think the readers will be able to find the plumes periods. With utmost respect to the reviewer's comment, we plan to keep the figure in its original format.

Figure 5 & 9: what are the vertical dashed lines? Are these marking specific plume periods? Please specify in the figure caption. Again, marking the different periods can make these types of figures more easy to digest. (See comment for Figure 3 & 8)

We changed Figure 5 as suggested by the reviewer, as below. However, in Figure 9, we have already used shading to indicate the solar radiation. Adding another shading may overcomplicate the figure. We thus leave this figure as is. We added "The vertical reference lines mark peaks of fire plumes." to the caption of Figure 9.

Figure 5 now appears as follows:

[Figure]

**Figure 5:** Time series of **(a)** backward trajectory cluster **(b)** hardwood primary BBOA markers and **(c)** softwood primary BBOA markers **(d)** general primary BBOA markers and **(e)** OC. Light red shadings show the plume periods, and light yellow shadings show the BB influence periods as described in Section 3.2.

Figure 7b: the colors between 3 and 4 and between 1 and 7 are difficult to distinguish. I recommend making all of the colors contrast more here; could also consider using different shapes to help with the contrast.

We changed those colors as suggested by the reviewer. We matched the colors in Figure 7 and 8. Now Figure 7 appears as:

[Figure]

[Figure]

**Technical comments:**

*There are a number of and minor grammatical errors (primarily in the form of word omissions; e.g. lines 32, 43-44, 223) throughout the paper.*

We rewrite the sentences as follows:

Line 32: These compounds have high $\overline{OS_c}$, and **they** are also semivolatile.

Line 43-44: Many gas phase volatile organic compounds (VOCs**), intermediate-volatile** and semi-volatile organic compounds (I/SVOCs) are also emitted in biomass fires (Grieshop et al., 2009a, 2009b; May et al., 2013).

Line 223: The concentration timelines were first normalized to prevent all the high (or low) concentration compounds **from being** clustered into the same factor.

*Line 257: 'The Oct 11 nighttime plume': this is referred to the Oct 12 plume in Fig 2. Please make sure how each plume is referred to (Oct 11, Oct 12, etc) is consistent throughout the paper.*

We changed the description to "**the October 11 night to October 12 early morning plume**", in both the writing and the caption of the figure. We also changed the first sentence in section 3.5 in the same way.

*Line 433-434: "The levels of PM2.5 at the sites considered here reached peak 0-5 hours after the Napa site." This sentence seems to be missing a word, such as "reached their peak"*

We thank the reviewer for pointing this out. We made this mistake 3 times in the article and we have all of them fixed.

---

## Author Comment (AC2) · 26 Feb 2021

**Response to Reviewer 2**

We thank the reviewer for the insightful comments and suggestions. We have addressed the reviewer's concern by making the following modifications to the paper.

General comments:

*The paper presents a wide range of topics and analyses. Little background is provided to bring the reader up-to-speed on the various analyses techniques or chemical mechanisms. I wonder if the paper should be split into two papers so more thorough discussions can be provided.*

We thank the reviewer for pointing this out.

Because the main text of this article is already quite lengthy, we moved some analytical details from this article into the Supplement and referred to other published articles for the analytical details and uncertainty information. Samples for this work were analyzed in the same batch with Jen et al. (2019), and the quantification standards used are also the same.

We respectfully prefer not to split this article into 2 at this stage. We are preparing another manuscript that focuses on the PTR-TOF-MS data from these same fire periods. Further discussion of the oxidation mechanisms will be provided in that paper.

*There is no mention of data quality or uncertainties in the paper. These are very low concentrations and especially when ratios are presented, I question the reliability of the values.*

We thank the author for this comment. However, in the main text, we have written that:

"As estimated by Jen et al. (2019), compounds exactly matched with a standard compound have an uncertainty ~ ±10%. Compounds quantified by the nearest compound in the same class have an uncertainty of ~ ±30%. Compounds with unknown functionality have a systematic uncertainty of 200%. We expect compounds with second column retention time > 1.6 s to also have such high uncertainty because there were no standard compounds with that high polarity, and a surrogate standard with lower polarity was used for quantification. However, only 7 reported compounds were in that chromatographic region with extremely high quantitative uncertainty."

We will add the following sentence to this paragraph.

Calibration was performed down to 2-10 ng for most compounds, and 20 ng for very polar compounds such as 2,4-dinitrophenol, 5-nitrovanillin and 4-nitrocatechol. In such concentrations, these analytes were observed at 10-10000 times the chromatographic signal-noise ratio. We can very conservatively assume the detection limit to be 1 ng. When we took a 3-hour sample at 21 lpm, the limit of detection was equivalent to ~ 0.26 ng m$^{-3}$. That is far below the concentrations of most compounds measured.

Specific comments:

*Line 33: Confusing sentence, suggest eliminating "primary and secondary" to simplify.*

We deleted "primary and secondary" as suggested by the reviewer. Now the sentence reads:

We observed no net particle-phase organic carbon formation, which indicates an approximate balance between the mass of evaporated primary and secondary organic carbonaceous compounds to the addition of secondary organic carbonaceous compounds.

*Line 135: Why does the temperature only go up to 320C?*

The temperature of the GC oven can go above 320°C. However, the thermal desorption system we have cannot ramp to temperatures far above 320°C. The GC oven final ramp temperature was therefore set at 320°C. Also, by ramping to 320°C and holding there for 5 minutes, compounds having volatilities close to C36 alkane can elute, defining the lower volatility range of observable compounds by this method which is typical of GC-MS analyses for biomass burning particulate matter.

*Lines 238-245: A comment on the accuracy of the forest inventories might be appropriate. I laughed at the 0.3% because these estimates have so much uncertainty.*

This 0.3% is obtained by spatially joining the fire perimeter and the post-fire canopy damage databased provided by the Sonoma County authority. They did not specifically report the uncertainty of this survey. We cannot comment on potential uncertainties from the survey but refer the reviewer to the report of that survey for the details (https://sonomaopenspace.egnyte.com/dl/sJcDLWK7U5/?). We agree that stating 0.3% here is unnecessary. We changed the sentence to:

"Conifer vegetation accounted for less than 1% of the area within the perimeter of the Atlas Fire."

*Line 270-271: This sentence seems out of place. I don't understand how it fits into the current paragraph. What is EI?*

We thank the reviewer for raising this issue. We changed the sentence to:

"Based on the structures, most of these sugars (if underivatized) can fragment into $C_2H_4O_2^+$ (*m/z* 60) under electron ionization, which can contribute to the signal of $C_2H_4O_2^+$ when they are measured by aerosol mass spectrometers (Fabbri et al., 2002)"

*Lines 279-288: This seems like introductory material since it is primarily on previous studies. Lines 288-289: This seems out of place, like a discussion or conclusion point.*
We agree that some sentences here do not fit very well into the result and discussion section. However, the description of the toxicity of PAHs and other compounds here can support our argument that evaluation of more compounds' health effects in the fire aerosols is necessary. We shortened this paragraph to:

"In all samples, the fraction of PAHs remained below 0.3% of total quantified OA. The low PAH fraction in OA measured at UCB could be a result of both low PAH emission and photochemical loss. The emission, exposure, and health impacts of PAHs in biomass burning received a great deal

of attention in previous studies (Shen et al., 2013; Sun et al., 2018; Tuet et al., 2019). However, other groups of compounds, such as monocyclic aromatic compounds including hydroquinone, catechol and cinnamaldehyde (Leanderson and Tagesson, 1990; Muthumalage et al., 2018), may also make substantial contributions to the toxicity of biomass burning smoke. The maximum concentrations of the most abundant positively identified compounds and most abundant PAHs observed at Berkeley and their possible hazards are listed in Table S1. Knowledge of the health impacts of inhaling compounds in this list are still lacking. For example, many nitroaromatic compounds were found to be mutagenic (Purohit and Basu, 2000). The nitro-compounds were found to be the main contributor to the mutagenicity of $PM_{2.5}$ in Northern Italy (Traversi et al., 2009). The sum of concentrations of (methyl-)nitrocatechols observed at Berkeley exceeded 1.2 µg m-3. However, no toxicological research of these compounds was found in PubChem."

*Line 326-327: Suggest rewrite of this sentence, as written it sounds like this work showed that hydroquinone and two other compounds were shown to be good tracers, but that's not the case.*

We changed the sentence to:

"Hydroquinone and two other compounds were also shown to be good tracers for manzanita burning in our previous work (Jen et al., 2018). They were present in most of the plumes, which is in line with our finding that manzanita is widely distributed in that region."

*Line 336: Suggest replacing "reached peak" with "peaked" for clarity.*

We changed it as suggested by the reviewer.

*Line 339-340: This is a strong statement for what looks like a weak pattern in the figure. Suggest softening to something like "our observations were consistent with the hypothesis that 7-oxo-DHAA to DHAA ratio is a useful indicator…"*

We changed this sentence as recommended by the reviewer. Now the sentence reads:

"Our observations were consistent with the hypothesis that 7-oxo-DHAA to DHAA ratio is a useful indicator for the formation of BB SOA."

*Line 354: Suggest adding a sentence addressing the diel concentration differences expected for OH and NO3 to complete this argument.*

We rewrote this part and added an important and recent reference by Palm et al. (2020) to substantiate our conclusion.

"Finewax et al. (2018) has shown that the molar yield of 4-nitrocatehcol when catechol reacts with OH and $NO_3$ are 0.3 ± 0.03 and 0.91 ± 0.06, respectively. **The dominance of $NO_3$ radicals as the nighttime oxidants can help to explain the higher nitrocatechols/OC ratio at night. It was recently reported that these nitrocatechols can be further oxidized by OH radicals, the major daytime oxidant (Palm et al., 2020).** The difference in oxidation mechanism is thus a more plausible explanation for the diel changes in the nitrocatechols/OC ratio **observed at Berkeley**."

*Line 450: "OSc of compounds can be satisfactorily predicted" This statement was not made in the previous sections, and I'm not clear on how you know it is "satisfactory".*

Although we did not explicitly draw this conclusion, we mentioned in line 206-207 that "The saturation vapor pressure model and the $\overline{OS_c}$ model achieved an $R^2$ of 0.93 and 0.96 between the modeled responses and the true responses, respectively (Figure S2)." In Line 202-295, we also mentioned the similarity of our prediction with Hatch et al (2018). But we understand that the word "satisfactory" may cause some concerns. We removed this word from the sentence. The revised sentence reads:

"We demonstrated that using easily obtained parameters from GC × GC measurements, the volatility and $\overline{OS_c}$ of compounds can be predicted."

*Line 734: Instead of "label", I suggest using "symbol" in the caption*

We thank the reviewer for this suggestion. We replaced the word "label" with "symbol".

*Figure 2: It is hard to distinguish the colors in these figures.*

The figure was changed to make the colors contrast more. We also changed two lines to dashed lines to make it easier to read. Now the figure appears as follows:

[Figure]

*Figure 3: It would be easier to digest if the legend was in the same order as the stacked bars.*

We thank the reviewer for this suggestion. We changed both Figure 3 and Figure 8 accordingly.

**Figure 3**

[Figure]

**Figure 8**

[Figure]